# Sodium Intake and Heart Failure

**DOI:** 10.3390/ijms21249474

**Published:** 2020-12-13

**Authors:** Yash Patel, Jacob Joseph

**Affiliations:** 1Lifespan Cardiovascular Institute, Warren Alpert Medical School at Brown University, Providence, RI 02914, USA; dryashpatel21@gmail.com; 2Department of Medicine, Veterans Affairs Boston Healthcare System, Boston, MA 02132, USA; 3Department of Medicine, Brigham and Women’s Hospital, Harvard Medical School, Boston, MA 02115, USA

**Keywords:** sodium, salt, heart failure, ambulatory heart failure, epidemiological studies

## Abstract

Sodium is an essential mineral and nutrient used in dietary practices across the world and is important to maintain proper blood volume and blood pressure. A high sodium diet is associated with increased expression of β—myosin heavy chain, decreased expression of α/β—myosin heavy chain, increased myocyte enhancer factor 2/nuclear factor of activated T cell transcriptional activity, and increased salt-inducible kinase 1 expression, which leads to alteration in myocardial mechanical performance. A high sodium diet is also associated with alterations in various proteins responsible for calcium homeostasis and myocardial contractility. Excessive sodium intake is associated with the development of a variety of comorbidities including hypertension, chronic kidney disease, stroke, and cardiovascular diseases. While the American College of Cardiology/American Heart Association/Heart Failure Society of America guidelines recommend limiting sodium intake to both prevent and manage heart failure, the evidence behind such recommendations is unclear. Our review article highlights evidence and underlying mechanisms favoring and contradicting limiting sodium intake in heart failure.

## 1. Salt and Sodium

Salt is an ionic compound made up of cation and anion. Edible salt consists of 40% sodium and 60% chloride by weight. Salt was historically used as a preservative since bacteria cannot flourish in the presence of high salt concentrations. Human cells require approximately 0.5 g/day of sodium to maintain vital functions. Most food preservatives have high sodium content and are major causes of increased dietary intake of sodium. The average sodium intake in most Americans is 3.4 g/day or 1.5 teaspoons of salt, which is greater than the physiological requirement for the human body. High sodium or salt intake can lead to chronic comorbidities including hypertension, heart failure (HF), chronic kidney disease, stroke, cardiovascular diseases, and increase mortality. Hence, current guidelines recommend restricting sodium consumption to 2–3 g/day [1].

HF is a major burden of morbidity and mortality on the health care system and is classified into two major groups, heart failure with reduced ejection fraction (HFrEF) and heart failure with preserved ejection fraction (HFpEF). Treatment of HFrEF involves both pharmacologic and non-pharmacologic strategies, while mainly heart rate and blood pressure control strategies are used in HFpEF since multiple clinical trials have not shown significant benefits of pharmacologic therapy [2]. Sodium restriction has historically been taught in textbooks as a cornerstone of the management of HF patients. However, data on this management strategy are controversial. In addition, the adherence to following a low sodium diet is challenging, especially after a recent hospitalization, as shown by Riegel et al. [3]. Before we vigorously start educating HF patients to limit sodium intake in their diet, we need to understand the evidence behind such recommendations. In this paper, we review evidence relating sodium to HF, pathophysiological mechanisms of increased sodium intake, and the relation of sodium intake to HF outcomes.

## 2. Guideline Recommendations for Sodium Intake

A low sodium diet is recommended in most national and international guidelines, as described in Table 1, with the intent of promoting health and preventing and managing comorbidities including HF.

## 3. Low Sodium Intake and Prevention or Management of HF

### 3.1. Evidence in Favor of Low Sodium Intake in Prevention or Management of HF

Systemic hypertension is one of the main risk factors for the development of HF. The lifetime risk of HF decreases with adequate treatment of blood pressure. Data from meta-analysis suggest a dose–response relationship between salt intake and increased blood pressure [12]. In a pooled analysis from four large prospective studies involving 133,118 patients, higher sodium intake was associated with increased risk of cardiovascular events and death compared with moderate sodium intake in hypertensive populations over a median of 4.2 years [13]. Systemic hypertension, if untreated, is a major risk factor for development of left ventricular hypertrophy. In the hypertensive patient population, diastolic dysfunction, left ventricular hypertrophy, and arterial stiffness are associated with urinary sodium excretion, and limiting sodium intake is associated with regression of left ventricular hypertrophy [14,15,16,17]. The proposed mechanism of regression of left ventricular hypertrophy with sodium restriction is improved large-arterial stiffness and microvascular endothelial dysfunction [18,19]. Sodium restriction is appropriate in patients with stage A (at risk for HF) and B (asymptomatic) HF due to its effect on lowering blood pressure, the incidence of hypertension, left ventricular hypertrophy, cardiovascular disease, and even incidence of HF [17,20,21,22,23,24]. However, there is insufficient evidence for such recommendation for stage C (with prior or current symptoms) and D (refractory) HF [25]. The Dietary Approaches to Stop Hypertension (DASH) diet, which emphasizes limiting sodium intake, has been shown to be associated with a lower incidence of HF in a prospective observational study of 36,019 participants in the Swedish Mammography Cohort over a course of seven years [26].

### 3.2. Pathogenic Mechanisms for Beneficial Effect of Low Sodium Intake in Management of HF

Figure 1 shows potential mechanisms of benefit with low sodium intake in patients with HF. A low sodium diet is shown to be associated with decreased pulmonary artery and capillary wedge pressures in patients with New York Heart Association (NYHA) Class III to IV heart failure [27]. Previous studies have shown that HF patients have systemic inflammation characterized by increased levels of tumor necrosis factor *(TNF)-alpha*, interleukin *(IL)-1B* and *IL-6*, chemokine (monocytes chemoattractant protein-1 and IL-8), as well as enhanced expression of adhesion molecules. Moderate sodium restriction (up to 2.8 g/d) was associated with reduced values of neurohormonal (B-type natriuretic peptide (BNP), aldosterone, plasma renin activity) and cytokine levels (*TNF-alpha, IL-6*) and increased levels of anti-inflammatory cytokine (*IL-10*) over 12 months of follow up compared to low sodium restriction (up to 1.8 g/d) [28]. A recent review of the effects of low dietary sodium intake in patients with HF revealed that 2.6–3 g/d of dietary sodium restriction is effective for decreased BNP, renin, and aldosterone plasma levels [29]. Similarly, low sodium intake in the DASH diet is associated with low systolic and diastolic blood pressure, arterial stiffness, and markers of oxidative stress including urinary F2-isiprostane levels in HFpEF patients [30]. Adherence to the DASH diet was shown to be associated with improvement in arterial compliance, improved exercise capacity, and quality of life in patients with stage C HF [31].

## 4. Low Sodium Intake and Worsening of HF

### 4.1. Evidence Against Low Sodium Intake in HF

In a randomized clinical trial, Aliti et al. studied the effect of the intervention of <2 g/d of salt intake in patients admitted with acute decompensated HFrEF with EF ≤ 45% on HF clinical congestion score compared to a control group with >2 g/d of salt intake [32]. On 30 days follow-up, there were no differences between the groups in the number of hospital readmissions and length of stay, though the patients in the intervention group had significantly more congestion than the control group (*p* = 0.02) [32]. Similarly, Velloso et al. did not see any significant difference in time needed for resolution of HF symptoms in adult patients admitted to the hospital with acute illness due to underlying chronic HF between the intervention group with <2 g/d salt intake and the control group with more than 2 g/d salt intake [33]. In a large Italian study in patients admitted with HF, patients assigned to low sodium intake (1.84 g/d) compared to moderate sodium intake (2.76 g/d), had reduced diuresis, more HF readmissions, poorer renal function, and a trend towards increased mortality [34]. Subjects in this study did not receive optimal neurohormonal blockade and received strict fluid restriction of 1 L/d and had high diuretic doses (up to 100 to 1000 mg of furosemide) without adjustment of clinical status. A recent pilot study done to see the effects of three-months of 1.5 g versus 3.0 g daily sodium intake in patients with HFrEF showed that both dietary interventions reduced urinary sodium without adverse quality of life improvements [35].

In animal models, sodium restriction in early stages of HF was seen to be associated with early aldosterone activation compared to normal or excess sodium intake [36]. These findings suggest that sodium restriction in early stages of HF should be avoided to prevent neuroendocrine disease progression. The data on sodium and fluid restriction in HFpEF patients are limited. A randomized clinical trial to see the effect of a diet with sodium and fluid restriction compared to an unrestricted diet in patients admitted with acute decompensated HFpEF showed that aggressive sodium and fluid restriction does not decrease readmission and mortality rate, and that it impairs the patient’s food intake without any significant neurohormonal effect [37]. A recent systematic review by Mahtani et al. in 2018 including nine randomized control trials that enrolled a total of 479 patients from a total of 2655 retrieved references, revealed no robust high-quality evidence of the effects of sodium restriction in patients with HF [38]. There was a trend in improvement of HF functional class symptoms in outpatient studies with reduced sodium intake, but no effects were observed on all-cause mortality, hospitalization, or length of stay [38]. Similarly, a recent randomized trial of 44 patients hospitalized for acute decompensated HF showed that a normal sodium diet (7 g/d), when compared to a low sodium diet (3 g/d) is associated with similar degrees of decongestion with lower neurohormonal activation during acute HF treatment [39].

### 4.2. Potential Mechanism for Adverse Impact of Low Sodium Intake in HF

Figure 2 shows the potential mechanism for decompensated HF with low sodium intake. In short, HF is characterized by activation of the sympathetic system and renin–angiotensin–aldosterone system (RAAS) activation due to decreased renal perfusion leading to sodium and water reabsorption from renal tubules [40,41]. A sodium-restricted diet in HF patients has been shown to be associated with activation of antidiuretic and anti-natriuretic systems [42]. A recent Cochrane review of 185 clinical studies randomizing persons to low- vs. high-sodium diet revealed that in plasma or serum, there was a statistically significant increase in renin, aldosterone, noradrenaline, adrenaline, cholesterol, and triglyceride levels in groups with low sodium intake as compared to groups with high sodium intake [43]. These increases in hormones can lead to further development of congestive symptoms. Vascular congestion in HF activates pro-oxidant and pro-inflammatory genes in endothelial cells, which contributes to cardiorenal dysfunction [44,45,46]. Reduced sodium intake can lower blood pressure, which in turn can increase the heart rate and thereby negate the effects of beta-blockers. This was shown in a recent meta-analysis of 63 studies, although the effect was marginal with a heart rate increase of as little as 2.4% [47]. Reverse causation could also explain the observed association of lower sodium intake and outcomes. Higher-risk individuals with HF might consume less sodium due to their underlying illness but still have higher risks of adverse events.

## 5. Potential Molecular Mechanism of Salt Diet and Heart Failure

The myosin heavy chain (*MHC*) protein is formed of *α* and *β* filaments. Changes in the proportion of these protein filaments are associated with cardiac mechanical performance. A high sodium diet is associated with an increase in cardiac expression of *β-MHC* and a decrease in the *α/β-MHC* ratio [48]. A low sodium diet was seen to be associated with increased *α/β-MHC* ratio, which in turn improves myocardial mechanical performance [48]. Similar effects of a high sodium diet were seen to be associated with myocyte enhancer factor (*MEF*) 2/nuclear factor of activated T cell (*NFAT*) transcriptional activity, and thereby increasing the expression of *MHC* genes [49]. Systemic hypertension can lead to a shift in the isoform distribution towards overexpression of the *β-MHC* gene with simultaneous downregulation of the *α-MHC* gene [50,51,52]. A high salt diet is associated with an increase in salt-inducible kinase 1 expression, which mediates the activation of *MEF2/NFAT* and genes associated with left ventricular hypertrophy [49].

There are five main proteins that are involved with calcium homeostasis and myocardial contractility—l-type Ca^2+^ channel (*LTCC*), phospholamban (*PLB*), *SERCA2a,* Na^+^/Ca^2+^ exchanger (*NCX*), and ryanodine receptors (*RYR*). A high sodium diet is also associated with reduced expression of both *PLB* and *NCX*. Ca^2+^ handling is important to maintain myocardial performance. The *LTCC* plays an important role in action potential during systole. Once Ca^2+^ enters the myocardial cell, it activates *RYR*, which in turn triggers Ca^2+^ release from the sarcoplasmic reticulum. This increase in Ca^2+^ release is responsible for the activation of myocardial contraction during systole. During diastole, the opposite mechanism happens; Ca^2+^ is pumped back from the cytosol to the sarcoplasmic reticulum by *SERCA2a* and sarcolemmal *NCX-1*, which mediates regulation of Ca^2+^ and Na^+^ exchange and thereby maintains excitation–contraction coupling. Altered Ca^2+^ handing is an important pathophysiological mechanism by which preclinical HF develops. Salt restriction has been shown to be associated with decreased *LTCC* protein levels in the left ventricle, increased *PLB* expression, and reduced *NCX* levels. Combined, these mechanisms together decreases sarcoplasmic reticulum Ca^2+^ overload by having an inhibitory effect on *SRCA2a* activity, and thereby is associated with a decrease in the contractility index [53,54,55].

## 6. Sodium Intake and Ambulatory Heart Failure

Low-sodium diet recommendations not only apply to hospitalized patients but also to ambulatory patients to prevent acute worsening of symptoms. However, the evidence behind these recommendations is not conclusive. Alvelos et al. reported that in patients with chronic HFrEF with Ejection Fraction (EF) ≤40%, sodium restriction was not associated with improvement in NYHA functional class during 15-day follow-up [42]. Colin-Ramirez et al. in 2004 showed that in patients with HFrEF or HFpEF, 2.0–2.4 g/d of sodium restriction was associated with an improvement in NYHA functional class and less reported signs of HF on 6-months follow up [56]. However, Colin-Ramirez et al. in 2015 showed no significant difference in NYHA functional class between the intervention group with sodium restriction of 1.5 g/d in patients and the control group of moderate sodium intake of 2.4 g/d in patients with HFrEF and HFpEF who are on optimal medical therapy during 6-months follow up [57]. In a study by Philipson et al., sodium and fluid restriction of 2.3 g/d and 1500 mL/d respectively were associated with lower NYHA functional class and symptoms of edema in patients with a history of HF in NYHA classes II and IV over a 12-week follow-up [58]. Hummel et al. reported that 30-day readmissions were lower in the group with sodium restriction of 1.5 g/d in patients with a history of hypertension and recent admission or acute decompensated HF who are followed by discharge into the community [59]. However, they reported that the Kansas City Cardiomyopathy Questionnaire clinical summary score was not different between the two groups over 12 weeks of follow-up [59]. Amongst 123 ambulatory HFrEF patients from two outpatient HF clinics over a median follow-up of three years, higher sodium tertile was associated with a 39% increased risk for all-cause hospitalization and a 3.5-fold increase in risk for mortality [60]. A recent propensity-matched analysis from the HF Adherence and Retention Trial showed that sodium restriction to <2.5 g/d in NYHA class II/III HF patients is associated with a 72% higher risk of death or HF hospitalization compared to a higher sodium intake of >2.5 g/d, especially in patients not receiving therapy with renin–angiotensin antagonists with a hazard ratio of 5.23 [61]. However, sodium intake was determined from a food-frequency questionnaire, which is subject to recall bias.

## 7. Sodium Intake in Selected Patient Populations

Recent meta-analyses of randomized control trials of treatment of hypertension reveal that the older population, non-white population, and only study groups with blood pressure in the highest 25th percentile show a clinically significant drop in blood pressure with a low sodium diet [62,63]. The Prospective Urban Rural Epidemiology study data showed that an increase in dietary sodium intake is associated with worse cardiovascular morbidity and mortality in a population with high basal sodium intake [64]. A dietary sodium restriction in such a population should be efficacious. Moreover, dietary sodium restriction was not efficacious in a population with low basal serum intake [64]. Amongst The National Health and Nutrition Examination Survey I participants over an average of 19 years of follow-up, a higher intake of dietary sodium was shown to be a strong independent risk factor for HF in overweight men and women with a body mass index of ≥25 kg/m^2^ [24]. Such effect was not seen amongst adult U.S. men and women with a body mass index <25 kg/m^2^. It also appears that sodium restriction is more beneficial for patients with advanced heart failure symptoms. Amongst 302 patients with HF, greater than 3 g/d dietary sodium intake was found to be associated with a hazard ratio of 2.54 (95% CI 1.10–5.84) for cardiac event-free survival in patients with NYHA III/IV HF symptoms compared to a hazard ratio of 0.44 (95% CI 0.20–0.97) in patients with NYHA I/II HF symptoms [65]. These data suggest that sodium restriction should be applied in only such a targeted population to obtain a substantial benefit. A study by Dolanski et al. examined the association of cognitive decline and low-sodium dietary adherence in 339 HF patients [66]. Interestingly, cognitive decline was not associated with low sodium intake; higher socioeconomic status and higher body mass index was associated with higher sodium intake. Similarly, Creber et al. studied the predictors of high sodium excretion in patients with previously or currently symptomatic HF amongst 280 community-dwelling adults [67]. They found that concomitant obesity and diabetes, and intact instead of deprived cognitive function, were associated with higher odds of sodium excretion. Similarly, sodium consumption was evaluated in 305 outpatients with HFrEF after receiving education to follow a <2 g sodium diet [68]. The authors found that sodium consumption exceeded recommended amounts in men and those with higher body mass indexes. These findings narrate the importance of addressing such demographic discrepancies to target in clinical trials to evaluate clinical outcomes with sodium restriction.

## 8. Serum Sodium Values and HF

Research has shown that low serum sodium value (hyponatremia) is seen in about 20% of hospitalized patients with acute HF [69]. Serum sodium concentration is closely regulated by water homeostasis, which in turn is regulated by thirst, arginine vasopressin, and kidney function [70]. Hyponatremia can be caused by excessive water retention from neurohormonal activation as well as by negative sodium balance from loop diuretics and with a low sodium intake diet [39]. Serum sodium values can be used to prognosticate outcomes in both HFrEF and HFpEF. Low serum sodium is a risk factor for poor long-term outcomes in acute HF, regardless of ejection fraction [71]. The Organize Program to Initiate Lifesaving Treatment in Hospitalized Patients with Heart Failure registry (OPTIMIZE-HF) involving 48,612 patients recruited from 259 hospitals revealed that each 3 mmol/L drop in serum sodium values below 140 mmol/L in hospitalized patients is associated with a 19.5% increased risk of in-hospital mortality, 10% increased risk of mortality on follow-up, and 8% increase risk of death or rehospitalization on follow-up [69]. A meta-analysis of HF patients showed that low serum sodium values are associated with an increased risk of mortality [72]. We have previously shown in a national Veterans Affairs database study of 25,540 HFpEF patients that a J-shaped relationship is observed between serum sodium levels and a higher risk of number of days of HF hospitalizations and all-cause hospitalizations per year [73]. Such a relationship exists with baseline measurements of serum sodium levels at the time of diagnosis of HF as well as during longitudinal follow-up. Among 50,932 HFpEF patients with a median follow-up of 2.9 years, a J-shaped relationship was seen between serum sodium values and all-cause mortality, HF hospitalizations, and all-cause hospitalizations [74]. These data are further supported by the fact that the improvement of hyponatremia in HF patients is associated with long-term clinical outcomes [75].

## 9. Future Directions

There are multiple clinical trials that aim to examine if sodium restriction in HF patients is associated with improved clinical outcomes. The Study of Dietary Intervention under 100 MMOL in Heart Failure (SODIUM-HF) is an open-label, multicenter, international, randomized controlled trial in ambulatory patients with chronic HF and aims to assess the effects of dietary sodium restriction on clinical outcomes [76]. The Geriatric out of Hospital Randomized Meal Trial in Heart Failure (GOURMET-HF) is a multicenter, randomized, single-blind, controlled trial of 3-months duration to see the effect of sodium restriction/DASH diet in older patients after discharge from acute decompensated HF admission [59].

## 10. Our Recommendations

There are likely many potential reasons for conflicting evidence regarding the benefit/harm of sodium restriction. These include heterogeneity of HF patient population studied, lack of uniformity in limiting the amount of sodium restriction per day, unclear data on associated use of fluid restriction, and simultaneous usage of diuretics and neurohormonal blockade agents. Given there is clear evidence of the benefit of limiting sodium intake to prevent various comorbidities leading to HF, we recommend limiting sodium intake in those who are at risk to develop comorbidities to prevent the onset of heart failure. In patients with HF, we recommend to continue limiting sodium intake to prevent morbidity associated with HF. We also recommend avoiding too much limitation in sodium intake as this has been associated with worse outcomes in HF patients.

## 11. Conclusions

The data supporting the restriction of dietary sodium intake in heart failure patients are unclear. While there appears to be a trend in reducing HF symptoms amongst patients using dietary sodium restriction, there appears to be no effect or slightly higher risk in mortality compared to no sodium restriction. A randomized control trial is hence needed to address this important clinical question.

## Figures and Tables

**Figure 1 ijms-21-09474-f001:**
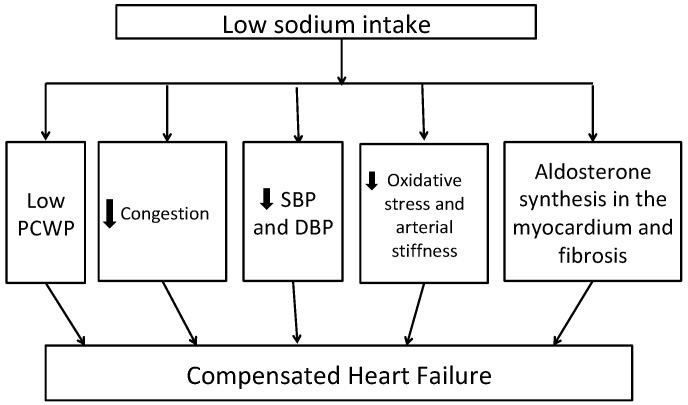
Potential mechanisms linking dietary sodium restriction to better heart failure outcomes DBP—diastolic blood pressure, PCWP—pulmonary capillary wedge pressure, SBP—systolic blood pressure. Abbreviations: DBP, diastolic blood pressure; SBP, systolic blood pressure; PCWP, pulmonary capillary wedge pressure.

**Figure 2 ijms-21-09474-f002:**
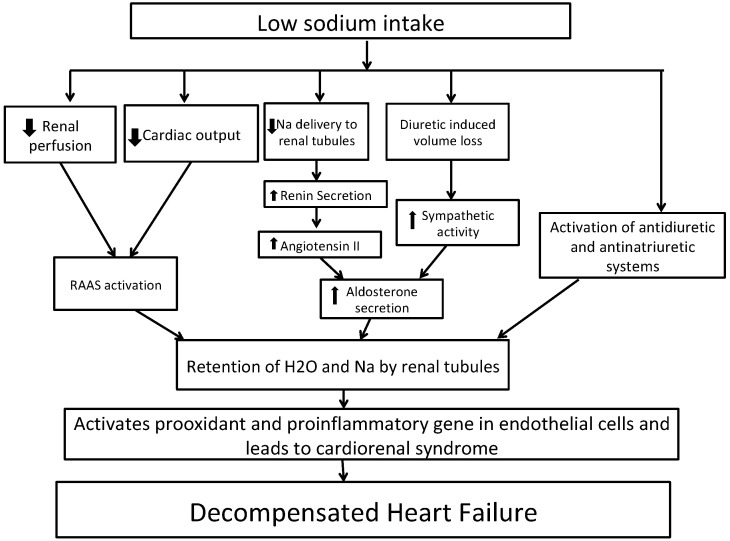
Potential mechanisms whereby dietary sodium restriction may worsen heart failure. Abbreviations: Na, sodium; RAAS, renin–angiotensin–aldosterone system.

**Table 1 ijms-21-09474-t001:** Guideline recommendations for sodium restriction in the general population.

Year, Name of Guideline	Sodium Restriction
2010, Dietary Guidelines for Americans [4]	<2.3 g/d in all adults <1.5 g/d in adults aged more than 50 years who are African American or with hypertension, diabetes, or chronic kidney disease
2013, World Health Organization [5]	<2 g/d in all adults
2020, American Heart Association [6]	<1.5 g/d in all adults
2010, Heart Failure Society of America [1]	2–3 g/d in all heart failure patients<2 g/d in patients with moderate to severe heart failure
2019, American Diabetic Association [7]	<2.3 g/d in patients with diabetes<1.5 g/d in patients with diabetes and hypertension
2016, European Society of Cardiology [8]	<5 g/d in all adults
2017, Canadian Cardiovascular Society [9]	<2 g/d in all adults
2015–2020 Dietary Guidelines for Americans [10]	2.3 g/d in all adults
2012, The Kidney disease: Improving Global Outcomes (KDIGO) [11]	<2 g/d in all patients with chronic disease not on dialysis

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
