# Peer review of "Sodium Intake and Heart Failure"

_ijms, 2020, doi:10.3390/ijms21249474_

Round 1

Reviewer 1 Report

This is a review article which discusses the benefit and harm of sodium restriction in the management of heart failure (HF). The Authors reported that even though Guidelines recommend sodium restriction, there are some conflicting evidence of benefit/harm of sodium restriction and discussed their potential mechanisms.

Major comments:

The Authors described the previously reported clinical or preclinical evidence. Please discuss, based on these evidence, what’s the potential mechanisms of conflicting evidence regarding the benefit/harm of sodium restriction. Also, please describe the final recommendation regarding the sodium intake.

Section 6. serum sodium level in HF is totally different topic from sodium intake/restriction. If the Authors want to mention something on this issue in this Review paper, please discuss the association of sodium intake/restriction with serum sodium level.

Minor comments:

Table 1. The listed Guidelines should be cited.

Author Response

We are very grateful for the reviewer’s comments and have addressed all the critiques to substantially improve the revised manuscript.

  1. This is a review article which discusses the benefit and harm of sodium restriction in the management of heart failure (HF). The Authors reported that even though Guidelines recommend sodium restriction, there are some conflicting evidence of benefit/harm of sodium restriction and discussed their potential mechanisms.

We thank the reviewer for these comments.

  1. The Authors described the previously reported clinical or preclinical evidence. Please discuss, based on these evidence, what’s the potential mechanisms of conflicting evidence regarding the benefit/harm of sodium restriction. Also, please describe the final recommendation regarding the sodium intake.

We appreciate the feedback on describing potential reasons for conflicting evidence of sodium restriction. We have described some of these reasons in updated manuscript in lines 299-301. As suggested, we have added a new section 10 describing our recommendations based on our review of the current literature.

  1. Serum sodium level in HF is totally different topic from sodium intake/restriction. If the Authors want to mention something on this issue in this Review paper, please discuss the association of sodium intake/restriction with serum sodium level.

Answer: We thank the reviewer for asking to clarify further the association of sodium restriction with serum sodium level before describing the association of low serum sodium values to HF outcomes. We have described these associations in lines 261-265

  1. The listed guidelines should be cited.

Answer: Thank you to the reviewers for pointing this out. We have included the references for various national and international guidelines in Table 1.            

Reviewer 2 Report

This is an interesting review manuscriptdescribing relationship between sodium intake and heart failure. It seems difficult to determine the optimal amount of salt intake in patients with heart failure.

I have two comments

  1. A pooled analysis of the results of four large studies, the PURE, EPIDREAM, ONTARGET, and TRANSCEND studies showed that salt restriction reduced the risk of cardiovascular events and death in populations with hypertension and that an association of extensively low sodium intake with increased risk of cardiovascular events and death is observed in those with or without hypertension (Mente et al. Lancet. 2016;388:465–75). I recommend the authors to cite the article.
  2. There are lots of typos, “?”in Figures.

Author Response

We are very grateful for the reviewer’s comments and have addressed all the critiques to substantially improve the revised manuscript.

  1. A pooled analysis of the results of four large studies, the PURE, EPIDREAM, ONTARGET, and TRANSCEND studies showed that salt restriction reduced the risk of cardiovascular events and death in populations with hypertension and that an association of extensively low sodium intake with increased risk of cardiovascular events and death is observed in those with or without hypertension (Mente et al. Lancet. 2016;388:465–75). I recommend the authors to cite the article.

Answer: We thank the reviewer for pointing out this important study that we inadvertently omitted. We have included this study and described the findings of the study in lines 67-70 of the revised manuscript.

  1. There are lots of typos, “?”in Figures.

Answer: We apologize for this oversight and have corrected typos in the manuscript itself. There are no “? ”in the Figures in our version of the manuscript. We have also added a list of abbreviations in Figure 1 which was lacking in the original submission.

Round 2

Reviewer 1 Report

The Authors properly responded to the Reviewer's comments. The Reviewer has no additional comments.